# Auditing Closed-Loop Learning in Recurrent Neural Networks:
# Reproduction, Robustness, and Generalization

## Abstract

Recurrent neural networks are often used as mechanistic models of learning and control, but closed-loop training creates reproducibility challenges because a model's actions alter future inputs. We conduct a claim-level reproducibility study of Ger and Barak's closed-loop RNN learning dynamics, testing independent implementation, seed variation, protocol perturbations, coupled-system diagnostics, and architecture/task transfer. Under a main-text-aligned double-integrator protocol, the trajectory-level peak, not a persistent final gap, reproduces strongly: 50/50 paired seeds show the post-initial open-loop deployed-loss peak, with mean peak/initial ratio 19.1, while final open-loop and closed-loop losses converge after open-loop recovery. The spectral stage and coupled-stability diagnostics also reproduce in 50/50 seeds. A targeted A1 analysis separates stability and behavioral tradeoffs: short-horizon improvements coincide with coupled-radius increases in 120/120 runs; long-horizon loss worsening occurs in 62/120, but never without the radius signal. Generalization is hierarchical: GRU and low-rank variants preserve final-loss divergence (the latter through deployed rollout blow-ups), tanh RNNs preserve the peak signature without a final gap, tracking transfer is strong, and path-integration final-loss transfer is weak. The peak signature and coupled-stability crossing persist across hidden sizes 50–400 with unchanged crossing epochs, while a two-layer stacked variant shifts to a persistent final gap without the strict peak. Local-Jacobian coupled spectra extend the stability diagnostic to gated controllers and ring tasks: crossings appear in 60/60 runs, decisively where behavioral transfer is strong and only marginally (final coupled radii near 1) for the weakly transferring partial-observation ring variants. These results motivate reporting practices for closed-loop RNN studies: deployed closed-loop loss, peak signatures, paired seeds, spectral stage criteria, coupled-system spectra, feedback strength, rollout horizon, and failure rates.

## 1 Introduction

Closed-loop recurrent neural networks occupy an important middle ground between supervised sequence modeling, control, and computational neuroscience. In a closed-loop environment, an RNN's output changes the future inputs it receives. This feedback makes the training problem qualitatively different from open-loop imitation or teacher-forced supervised learning, where future inputs are independent of the learner's deployed actions. Ger and Barak (Ger and Barak, 2025) recently argued that this distinction produces learning dynamics that are fundamentally different from those of otherwise identical open-loop RNNs. Their paper is valuable because it makes a mechanistic claim rather than only a performance claim: the coupled agent-environment system, not the RNN alone, determines important learning transitions.

This paper is not a figure-level reproduction study. We use Ger and Barak's framework as a test case for claim-level reproducibility in recurrent control systems, asking which conclusions survive independent implementation, seed variation, optimizer and hyperparameter perturbations, coupled-system stability analysis, and transfer to nonlinear or path-integration-style settings. A rerun of a public artifact can be educational,

but it is broadly useful only when it surfaces reusable lessons about when claims are reliable, when they are protocol-sensitive, and what future papers should report.

Our study makes four contributions: an independent, config-driven reproduction of the central open/closed-loop comparisons in Ger and Barak; quantitative, seed-level tests with confidence intervals for the qualitative stage and stability claims; robustness measurements under optimizer, initialization, feedback, noise, and horizon perturbations that separate stable from protocol-sensitive conclusions; and an audit checklist for future closed-loop RNN studies.

## 2 Related Work

**Task-trained RNNs in computational neuroscience.** Task-trained RNNs are widely used as mechanistic models because they solve cognitive and motor tasks while exposing internal dynamics for analysis: trained networks can implement low-dimensional dynamical mechanisms (Sussillo and Barak, 2013), account for context-dependent computation in prefrontal cortex (Mante et al., 2013), and serve as hypothesis-generating models in systems neuroscience (Barak, 2017). Large-scale analyses also show that architecture and training details change representational geometry even when task performance is similar (Maheswaranathan et al., 2019), which motivates our claim-level framing: reproducing a task curve is not enough if the mechanistic conclusion depends on protocol or model class.

**Dynamical-systems analysis of RNNs.** The original paper sits in a tradition of reverse-engineering trained RNNs through fixed points, eigenvalues, and linearized dynamics (Sussillo and Barak, 2013; Maheswaranathan et al., 2019). Our audit differs in emphasis: we ask not only whether a trained RNN has interpretable internal dynamics, but whether the relevant dynamical object is the RNN alone or the coupled agent-environment system—central in closed-loop settings, where the environment transforms actions into future observations.

**Open-loop training, closed-loop deployment, and distribution shift.** The mismatch between teacher-forced training and deployed rollouts is well known: scheduled sampling addresses it in sequence prediction (Bengio et al., 2015), and DAgger addresses the fact that a learner's actions change the future states it visits (Ross et al., 2011). Closed-loop RNN learning has an analogous but mechanistically richer mismatch—open-loop imitation can minimize teacher-forced error while failing under deployed rollouts—which motivates our insistence on reporting deployed closed-loop loss.

**Reproducibility and robustness in machine learning.** Our audit follows the broader shift from artifact reruns to structured reproducibility studies, from the NeurIPS reproducibility program's emphasis on artifacts and checklists (Pineau et al., 2021) to evidence that deep-RL conclusions can hinge on seeds, evaluation protocol, and implementation details (Henderson et al., 2018). Closed-loop RNN studies share these risks because outcomes can depend on initialization, rollout horizon, optimizer, and environment coupling; we therefore treat Ger and Barak's work as a case study for a reusable audit protocol.

We use *reproduction* for tests that preserve the original task family and claim, *robustness* for perturbations of seeds, optimizer, initialization, horizon, noise, feedback, and clipping, and *generalization* for architecture or task-family changes. This terminology is intentionally conservative: a robustness or generalization failure does not by itself falsify the original result, but it does delimit where the claim should be reported as reliable.

## 3 Original Claims and Audit Questions

We separate the original claims under reproduction from audit questions introduced by this study. The reproduced claims are C1–C3. C1 is the central divergence claim: identical RNNs trained open-loop and closed-loop follow different learning trajectories and differ under closed-loop deployment. C2 is the stage claim: closed-loop learning passes through distinct learning stages. C3 is the mechanistic stability claim: progress depends on stability of the coupled agent-environment system.

The audit questions are A1–A2. A1 asks whether C1–C3 are robust to protocol perturbations and whether a targeted short-vs-long horizon analysis reveals the expected stability tradeoff. A2 asks whether the phe-

nomena generalize across architectures and tasks. We use A1 and A2 to avoid implying that Ger and Barak made every robustness or generalization claim in exactly our framing.

For each reproduced claim or audit question, we define three evidence levels. A result is *reproduced* when the effect and diagnostic hold under the relevant protocol for most paired seeds, with confidence intervals excluding zero or a pre-specified smallest effect size of interest. For final-loss C1 diagnostics this smallest effect is a 5 percent relative deployed-loss gap, corresponding to roughly 0.022 absolute loss in the original-setting closed-loop scale; smaller final gaps are reported continuously but not counted as final-loss divergence. Appendix E varies this threshold from 1 to 20 percent: every setting keeps its classification except the partial-observation tanh ring, whose common sub-threshold gaps (19/20 seeds above 1 percent, 2/20 above 5) are exactly what the smallest-effect convention is meant to exclude. Because the original figure also emphasizes a transient open-loop peak, we report peak-over-training support as a separate C1 signature. A result is *partially reproduced* when the direction appears but is unstable across seeds, sensitive to plausible perturbations, or only weakly supported by the diagnostic. A result is *not reproduced* when the effect is absent, reverses, or depends on hand-picked visual criteria. Table 3 in Appendix C maps each original claim's evidence to our test and decision.

## 4 Closed-Loop RNN Reproducibility Audit Protocol

We specify a modest audit protocol rather than proposing a new benchmark. The protocol is a recipe for testing claims about closed-loop RNN learning dynamics in any recurrent control setting where actions can affect future observations. Its inputs are: a set of original claims, a task/environment, an open-loop training procedure, a closed-loop training procedure, a seed set, a small set of protocol perturbations, and at least one held-out architecture or task variant. Its outputs are: per-seed deployed closed-loop metrics, stage labels, stability diagnostics, perturbation summaries, generalization summaries, and a claim-level reproducibility matrix. In brief: paired training from cloned initializations, seed-level uncertainty with explicit counts and bootstrap intervals, detectors predefined in the claim's own diagnostic space, coupled agent-environment spectra wherever (local) linearization is possible, perturbations that alter optimization or coupling, at least one architecture and one task variant, and a uniform decision rule across settings. Table 5 in Appendix C gives the full specification.

The protocol deliberately separates *measurement* from *interpretation*. Measurement consists of paired runs, deployed losses, spectra, stage labels, and perturbation outcomes. Interpretation happens only after aggregating seed-level outputs into claim-level decisions. This separation is important because closed-loop RNN papers often make qualitative statements about stages or mechanisms from a small number of curves; our protocol requires those statements to be tied to explicit detectors, uncertainty estimates, and failure rates.

### 4.1 Metrics and run accounting

The main metrics are deployed closed-loop loss, teacher-forced imitation loss, effective feedback gain, coupled and RNN-only spectral radius, algorithmic stage labels, and A1 multi-horizon tradeoff components. Exact formulas are in Appendix B. The audit comprises 675 completed paired runs across a versioned audit suite: 50 core C1–C3 runs, 195 robustness runs, 120 targeted A1 multi-horizon runs, 90 A2 extension runs, and 220 runs for the scale/depth, clipping-selectivity, convention-ablation, local-Jacobian, and artifact side-by-side analyses. We additionally performed two one-seed protocol sanity checks and five seeded re-executions of the original nonlinear notebook. All reported tables and figures are generated from the accompanying code and saved outputs. Non-finite outcomes are classified as failures (F), not as zero-support evidence. Detector thresholds were fixed during smoke-test development, before any reported sweep was run, and were not revisited afterwards. To show that the conclusions are not artifacts of those choices, Appendix E varies the final-gap threshold across a $20\times$ band (1–20 percent) and the A1 improvement/worsening thresholds and horizon pairs across a grid; no claim-level decision changes at or above the pre-specified values, and the A1 union, radius, and exclusive loss-only outcomes are identical in every grid cell.

## 4.2 Paired open-loop and closed-loop training

For each seed, one initialization is cloned into an open-loop and a closed-loop learner. Following Ger and Barak's use of "policy gradients," the closed-loop controller is optimized by taking gradients of the rollout objective with respect to the RNN policy parameters. In this differentiable double-integrator setting, those gradients are computed by backpropagation through the unrolled agent-environment dynamics, matching the direct-gradient setup of the public artifact rather than REINFORCE-style stochastic likelihood-ratio estimation. The open-loop model imitates teacher trajectories in our independent implementation. Both are evaluated under deployed closed-loop rollout.

## 4.3 Stage detection

The original paper anchors its stages spectrally: Stage 1 is a negative-position policy with an unstable complex pair, Stage 2 ends when the dominant eigenvalues of the coupled matrix enter the unit disk, and Stage 3 involves policy refinement and growth of a third real mode. Our main C2 detector therefore operates in the same spectral space: it records the effective position gain, the unstable complex coupled mode, the coupled stability crossing, and post-crossing growth of the third real mode. We also report loss-only derivative and segmented-changepoint observers as downstream diagnostics, not as substitutes for the spectral claim.

## 4.4 Coupled-system stability diagnostics

For linearized settings, we compute the spectral radius of the RNN recurrent matrix and of the coupled agent-environment transition matrix. This targets C3 directly: if closed-loop dynamics are governed by the agent-environment loop, the coupled diagnostic should be more informative than an isolated RNN diagnostic.

## 4.5 Robustness and generalization

The A1 robustness sweep changes learning rate, initialization scale, feedback strength, episode length, observation noise, control penalty, optimizer, and gradient clipping. The Adam condition is included because Ger and Barak's Appendix C.7 predicts that adaptive optimization mitigates the structured short/long-horizon conflict. The A2 generalization sweep tests nonlinear RNNs, GRUs, low-rank RNNs, tracking control, and ring/path-integration-style tasks.

## 4.6 Original artifact, independent implementation, and deviations

We preserved the public artifact under `external/original_artifact/`. It contains the original notebooks for figure reproduction, nonlinear training, theory, and tracking, plus the pickle files used by the artifact. Our reproduction log records which files load, which notebooks depend on precomputed data, and which notebook runs are long or checkpoint-heavy.

The experiments reported here use an independent, config-driven implementation rather than direct notebook execution. The double-integrator task, paired open/closed training, effective gains, coupled spectra, stage detectors, robustness perturbations, and generalization tasks were reimplemented as package modules and command-line scripts. We matched the original double-integrator architecture and training profile where needed for C1–C3, then intentionally introduced new A1/A2 audits over seeds, optimizer, initialization, feedback, horizon, noise, clipping, architecture, and task. We did not contact the original authors; all choices were made from the paper, public artifact, and documented configs. Exact notebook reruns are therefore treated as artifact checks, while the paper's quantitative claims are evaluated through the independent implementation. Every table and figure can be regenerated from saved raw CSVs with `scripts/run_full_audit.sh`, `scripts/run_targeted_c2_a1_a2.sh`, and `python -m closed_loop_repro.plotting.make_all_figures -config configs/figures/tmlr.yaml`.

A code-level comparison shows the core reproduction matches the original double-integrator dynamics, hidden size, tanh controller class, SGD learning rate, batch size, rollout horizon, gradient clipping, and teacher-student design, though it is not a line-for-line reimplementation. The main-text-aligned config sets $\beta = 0$, samples $x_0 \sim [-2, 2]^2$, averages the squared Euclidean state cost over the rollout, and keeps the notebook's

timestep ordering. We therefore interpret the package as an independent claim-level audit of the original protocol family, not an exact numerical rerun of every notebook.

Appendix D makes this provenance systematic: every protocol setting is classified as an exact artifact match (22 settings), a main-text-aligned deviation from the public notebook (5: $\beta$, the two $x_0$ initialization bounds, the state-cost reduction, and the loss normalization), or a new audit choice (10: evaluation protocol, detector thresholds, perturbation grid, seed counts). Seeded re-executions of the original nonlinear training notebook agree with package runs under the notebook protocol on final losses, open-loop peak epoch, and peak ratio (Appendix H).

This protocol alignment changed the interpretation of C1. During development, an open-loop implementation that trained the student on teacher-rollout inputs rather than the paper/artifact white-noise convention produced a persistent final deployed-loss gap; after correcting the mismatch, the faithful protocol shows the original transient open-loop peak followed by final convergence. The mechanism is persistent excitation rather than a numerical detail: white-noise teacher inputs supervise the student across the whole observation space, so imitation eventually identifies the teacher's full input-to-action map and deployed losses converge (the peak appears while the partially identified policy transits deployment-destabilizing parameter regions), whereas teacher-rollout inputs concentrate supervision on the teacher's on-policy distribution, leaving off-distribution behavior uncorrected under deployment—the covariate-shift mechanism of imitation learning (Ross et al., 2011; Bengio et al., 2015). A controlled ablation under the core protocol confirms this reading: teacher-rollout inputs produce a persistent final deployed-loss gap in 15/15 seeds (median relative gap $\approx 147$) with no post-initial peak (0/15), while the white-noise convention shows the peak with final convergence in all 65 comparable runs (Appendix G). We therefore report C1 with two sub-diagnostics throughout: the peak-over-training signature and the final deployed-loss gap.

## 5 Result 1: Direct Reproduction of the Core Divergence

The original-setting double-integrator runs reproduce the trajectory-level C1 signature (Figure 1). Across 50 paired seeds, every open-loop student exhibits a post-initial deployed-loss peak. The peak occurs at mean epoch 249.1 with bootstrap 95 percent CI $[247.5, 250.5]$ and median epoch 249.5. The mean peak/initial ratio is 19.1 with CI $[17.1, 21.4]$, and the mean peak/final ratio is $4.88 \times 10^4$. This is the qualitative open-loop failure mode emphasized in the original figure: teacher-forced imitation can look successful while deployed closed-loop loss transiently explodes.

Final losses converge after the open-loop recovery. The closed-loop trained model reaches final deployed loss 0.4370 with bootstrap 95 percent CI $[0.4367, 0.4373]$, seed SD 0.00109, and IQR $[0.4361, 0.4378]$. The open-loop trained student has final loss 0.4387 with CI $[0.4383, 0.4390]$, seed SD 0.00126, and IQR $[0.4379, 0.4393]$. The paired final gap is positive but small, 0.00168 with CI $[0.00147, 0.00190]$, and does not exceed our 5 percent final-loss threshold in any seed. Thus C1 reproduces as a learning-trajectory divergence and peak-over-training signature, not as a persistent final deployed-loss separation.

The spectral C2 and C3 signatures also appear in the core run. The spectral three-stage detector is supported in 50/50 seeds, with median Stage 1 end at epoch 7 and median Stage 2/stability crossing at epoch 105.5. Coupled stability crossings occur in 50/50 seeds, and the final coupled spectral radius is 0.6253 with CI $[0.6243, 0.6264]$.

As a direct check on the independent implementation, we executed the original artifact's nonlinear training notebook with injected seeds and compared it against package runs under the notebook's own protocol: the two agree within seed variability on closed/open final losses, open-loop peak epoch, and peak/initial ratio, and the notebook itself shows the transient peak with final convergence (Appendix H).

## 6 Result 2: Robustness and Tradeoff Audit

The robustness sweep consists of 195 paired runs: 13 perturbation settings with 15 seeds per setting, covering learning rate, initialization scale, feedback strength, rollout horizon, observation noise, optimizer, control penalty, and gradient clipping (Table 4 in Appendix C; Figure 2). The peak-signature version of C1 is

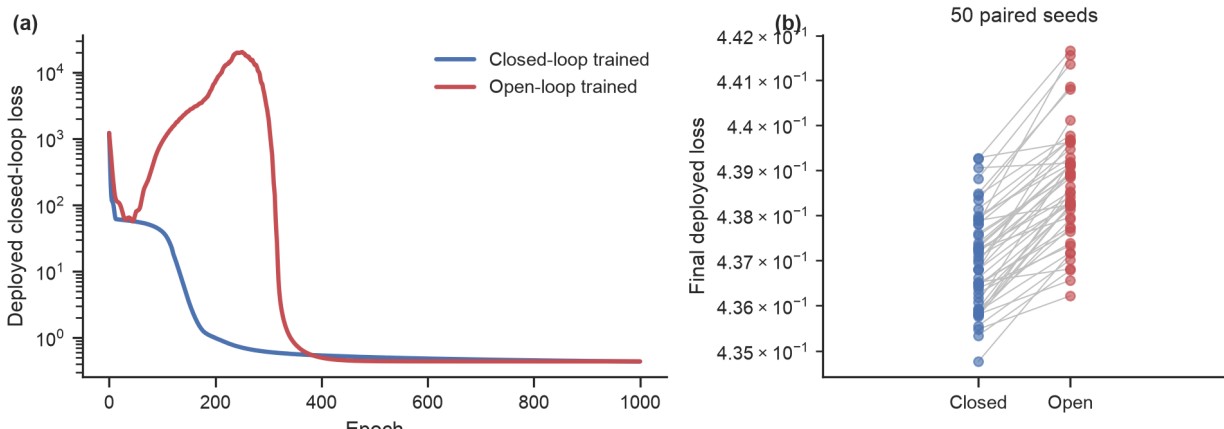

Figure 1: Core C1 reproduction under the original double-integrator setting. (a) Median deployed closed-loop test loss across paired seeds, with interquartile bands. The open-loop student shows the post-initial deployed-loss peak in 50/50 seeds. (b) Paired final deployed losses for the same initializations. Final losses are similar after open-loop recovery, so the reproduced C1 evidence is the trajectory-level peak rather than a large final gap.

robust across finite SGD-like perturbations: it appears in 165/195 robustness runs, i.e. all finite settings except Adam. The final-loss version is more selective, supported in 60/195 runs. Those 60 successes are exactly the low-learning-rate, high-learning-rate, short-horizon, and Adam rows (15/15 each). The baseline robustness row is 0/15 for final-loss C1, matching the 0/50 core run; long horizon, feedback, initialization, noise, and high-control-penalty rows are also 0/15 because open-loop recovery erases the final gap. Adam is therefore not a simple weakening of all closed/open signatures: it removes the transient peak (0/15) but leaves a persistent final gap (15/15)—an optimizer-specific failure mode rather than SGD-like peak-and-recovery. With gradient clipping disabled, all 15 `no_clip` runs produce non-finite final losses and are counted as failures rather than negative evidence for the claim. Two follow-ups localize this failure (Appendix F). Forensically, the closed-loop learner diverges first in all 15 seeds—non-finite training loss by epoch 5–7 from an initial coupled radius at or marginally above the stability boundary (1.000–1.045)—and, as the imitation teacher, poisons the open-loop student from its first epoch. A selectivity sweep then shows the failure is specific to the closed-loop learner and the SGD regime: clipping only the closed loop keeps 15/15 runs finite with the usual peak-and-recovery, an unclipped closed loop fails 15/15 regardless of the open loop, unclipped Adam survives 15/15, and unclipped SGD at a 3× lower learning rate still fails 15/15.

The stage result splits by diagnostic (Figure 7 in Appendix E). The spectral C2 detector is supported in 180/195 robustness runs, failing only when training becomes non-finite without clipping. Loss-only stage detectors are more diagnostic-dependent: the derivative-style loss detector finds three-stage structure in many finite runs, but segmented changepoints on the original loss curves do not recover the same boundaries. This supports the reporting lesson that stage claims should be tied to the diagnostic space in which they are defined.

The coupled-system stability transition is robust wherever the coupled spectrum is available and the run remains finite. It is supported in 180/195 robustness runs, again failing only in the no-clipping condition. This supports the original mechanistic emphasis in linearizable settings and separates spectral stability from downstream loss-changepoint sensitivity.

The targeted A1 sweep is more specific than the original union metric suggests (Figure 3). Across six control penalties and 20 seeds per penalty (evaluation horizons $T_s = 10$ and $T_\ell = 200$; Appendix B), the union criterion–short-horizon improvement plus either long-horizon loss worsening or increased coupled spectral radius–is supported in 120/120 runs, with mean conditional fraction 0.4724. Splitting the criterion shows that the radius signal is universal: short-horizon improvement plus increased coupled radius is supported

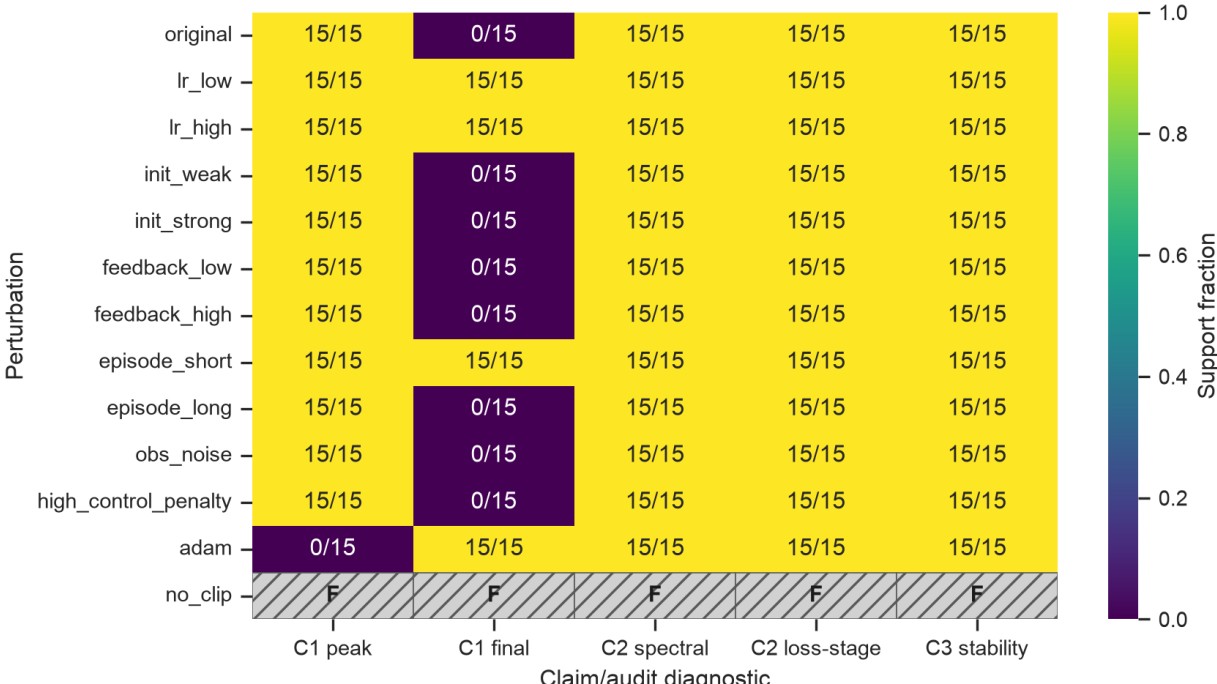

Figure 2: Robustness heatmap over perturbation settings and claim diagnostics. C1 is split into the transient peak signature and the final deployed-loss criterion, so the original row correctly shows peak support but no final-gap support. C2 is split into spectral support and a broader downstream loss-stage diagnostic. A1 is omitted here because the coarse robustness proxy is not the targeted multi-horizon A1 test in Figure 3. Disabling gradient clipping produces non-finite failures marked as F rather than zero support.

in $120/120$ runs. Long-horizon loss worsening occurs in $62/120$ runs, and the intersection of loss worsening and radius increase is also $62/120$. The exclusive loss-only criterion is $0/120$, while the exclusive radius-only criterion is $120/120$. This distinction prevents the audit from overstating a purely behavioral loss tradeoff: every detected loss-worsening case is accompanied by the stability-cost signal. The converse does not hold. In $58/120$ runs the radius increase appears without long-horizon loss worsening, so the coupled-radius signal is a necessary accompaniment of the behavioral tradeoff but not a sufficient predictor of it: the coupled system can absorb a stability cost without a measurable behavioral cost at the evaluated horizon. Radius-only cases concentrate at high control penalties. With 20 seeds per penalty, the radius-only counts are 5, 5, and 6 at $\beta = 0, 0.001$, and $0.005$, rising to 13, 13, and 16 at $\beta = 0.02, 0.05$, and $0.1$ (totalling $58/120$; the complementary 62 are loss-worsening), while median final coupled radii are nearly identical across the two classes—stronger control penalties damp the behavioral consequence of the stability cost without removing the stability signal itself. These A1 conclusions are also stable under detector variation: the union and radius criteria remain $120/120$ and the exclusive loss-only criterion remains $0/120$ in every cell of a grid over $\delta \in \{0.0025, 0.005, 0.01, 0.02\}$ and horizon pairs $\{(10, 200), (10, 50), (50, 200)\}$ (Appendix E).

## 7 Result 3: Which Diagnostics are Predictive?

The central C2 result is diagnostic-specific. The spectral detector reproduces the original stage structure in $50/50$ core seeds: negative position gain early, an unstable complex coupled mode, dominant coupled eigenvalues entering the unit disk near epoch 105.5, and subsequent growth of the third real mode. In contrast, segmented regression on deployed loss alone finds median changepoint boundaries at epochs 120 and 193 with slopes $-0.0117, -0.0432$, and $-0.00065$. This loss-only observer does not recover the original stage ordering under any detector variant: strict three-stage support remains $0/50$ across looser and stricter

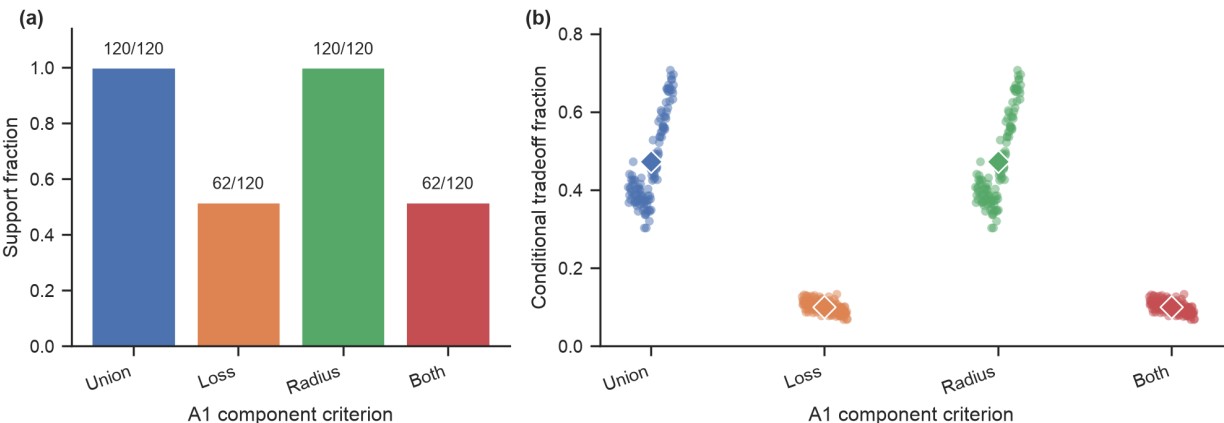

Figure 3: Targeted A1 component analysis over 120 paired tradeoff runs. (a) Support counts for the union criterion, long-horizon loss-worsening criterion, radius criterion, and the intersection of loss and radius criteria. (b) Seed-level conditional fractions among short-horizon improvement intervals. The union and radius criteria are supported in 120/120 runs; loss worsening appears in 62/120 and never without the radius signal. Diamonds show means.

thresholds and across variants that perturb the segmentation itself (Table 9 in Appendix E); the fitted boundaries are insensitive to the labeling thresholds because the underlying least-squares segmentation does not depend on them. Thus C2 is reproduced spectrally, while loss-only visual stage boundaries are not a reliable substitute for spectral criteria.

The original-setting coupled-system stability transition is detected in 50/50 seeds (Figure 4). The final coupled spectral radius is finite and below the stability threshold in every original run, and the median crossing epoch is 105.5. Because the spectral C2 detector defines the Stage 2 boundary by this crossing, the substantive timing test is whether the crossing aligns with the learning plateau exit. It does under the derivative-style plateau detector: the median stability-to-plateau gap is one epoch. Segmented loss-only changepoints give different downstream boundaries, reinforcing that C3 should be evaluated in coupled spectral space rather than from visual loss-curve annotations alone.

The spectral diagnostic now covers both explicitly constructible and locally linearized systems. For linear, tanh, low-rank, and stacked controllers on linear tasks, the coupled transition matrix is constructed in closed form. For gated controllers and nonlinear tasks, where no closed form exists, we compute local Jacobian spectra: the Jacobian of the one-step coupled map $(s_t, h_t) \mapsto (s_{t+1}, h_{t+1})$, evaluated by automatic differentiation at points along deployed rollouts. The two constructions validate each other where both apply: on linear and tanh double-integrator systems, the Jacobian spectral radius at the origin matches the closed-form coupled radius to within $10^{-5}$, and on 10 core-protocol seeds instrumented with both constructions, the along-trajectory Jacobian radius tracks the zero-point closed-form radius with median absolute difference $3.6 \times 10^{-7}$ and correlation 1.00 (Appendix I). Tracking uses a different coupled-matrix construction from the double integrator, which makes its 3/10 C3 support a weaker diagnostic transfer result. GRU and ring diagnostic-transfer results from the Jacobian construction are reported in Section 8.2.

# 8 Result 4: Does the Phenomenon Generalize?

## 8.1 Behavioral transfer

Generalization of the deployed-loss and peak signatures is hierarchical rather than uniform (Figure 6). Across 90 architecture/task extension runs, final deployed-loss divergence is supported in 38/90 runs, while the open-loop peak signature appears in 60/90. Architecture transfer within the double-integrator-style control family depends on the diagnostic: GRU variants show final-loss divergence in 10/10 seeds, and low-rank

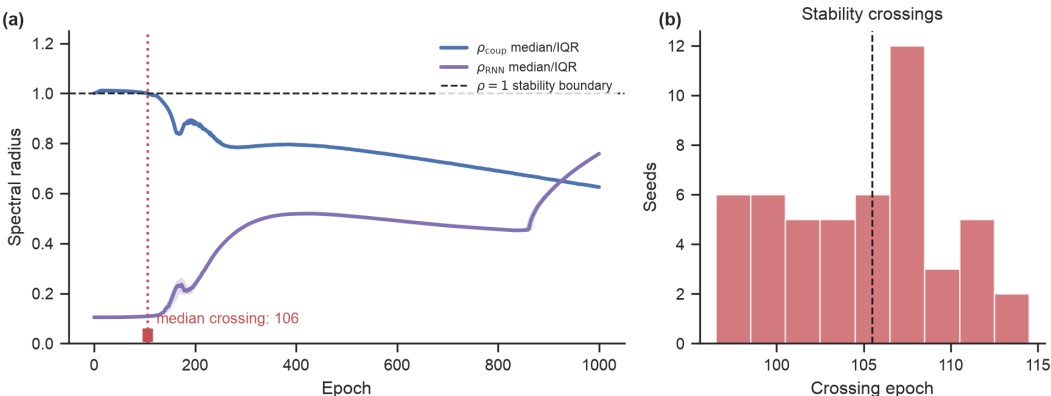

Figure 4: Coupled-system spectral analysis for the original setting. (a) Median coupled and RNN-only spectral radii across 50 seeds, with interquartile bands; the dashed line marks the $\rho = 1$ stability boundary and tick marks show per-seed crossing epochs. The plotted median marker is rounded to epoch 106; the unrounded median used in the text is 105.5. (b) Distribution of stability-crossing epochs. The coupled spectral radius crosses below the stability boundary early in training, whereas the RNN-only spectral radius remains far from the relevant stability threshold.

variants also pass in 10/10 seeds but through open-loop deployed rollout blow-ups rather than calibrated gaps (Figure 6, clipped and marked with triangles). Tanh RNNs preserve the peak signature in 10/10 but converge to nearly identical final losses. Task transfer to tracking is strong under the final-loss criterion, with deployed-loss divergence in 10/10 seeds. Task-family transfer to the harder partial-observation moving-target ring variants is weak on the deployed-loss criterion, with 5/20 GRU and 2/20 tanh seeds passing; the open-loop peak, by contrast, appears in 20/20 GRU and 0/20 tanh seeds, so for the partial-observation GRU the peak signature transfers cleanly while the final gap does not.

The simple ring variant is best viewed as a sanity check, not a hard generalization test: with a fixed target the mapping is nearly memoryless, and as expected it has essentially no supporting seeds (1/10). The harder ring follow-up is fairer but still weak (mean gaps 0.044 GRU, 0.051 tanh). The closed/open divergence thus transfers across some architecture and tracking changes but not automatically to path-integration variants.

## 8.2 Diagnostic (C3) transfer

Diagnostic transfer is a separate question from behavioral transfer, and we report it separately: a variant can preserve the deployed-loss divergence while the stability mechanism remains unmeasured, and vice versa. Under the closed-form construction, the C3 stability transition is supported in 10/10 tanh RNN runs and 10/10 low-rank runs, and in 3/10 tracking runs, with the caveat that tracking uses a different coupled-matrix construction than the double-integrator case. For GRU and ring variants, where no closed form exists, we use the local Jacobian construction of Section 7, under the separate flag `claim_C3_stability_transition_jacobian` so the two evidence types are never pooled. Crossings are detected in 60/60 Jacobian runs (GRU double integrator 10/10, ring 10/10, partial-observation ring 20/20 for both controllers), but the depth of stabilization tracks behavioral transfer: the GRU double integrator and simple ring stabilize decisively (median final coupled radii 0.58 and 0.83; crossings at epochs 10 and 5), while both partial-observation variants stay near-marginal throughout training (final radii 0.986 and 0.978, seeds up to 1.19; crossings at epochs 40 and 270)—precisely where behavioral transfer is weak (Appendix I). The reporting lesson stands: distinguish performance transfer from diagnostic transfer.

## 8.3 Scale and depth

All experiments above use single-layer controllers with hidden size 100 (GRU 64). To test whether the audited signatures are artifacts of that scale, we repeat the paired core protocol at hidden sizes 50, 200,

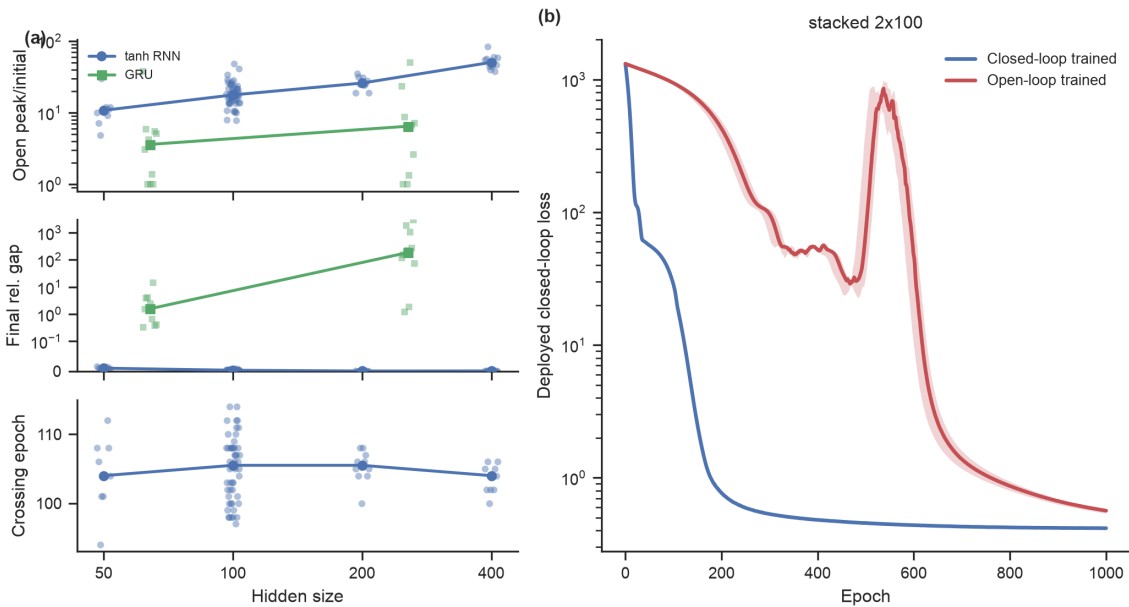

Figure 5: Scale and depth. (a) Peak/initial ratio, final relative gap, and stability-crossing epoch versus hidden size (10 paired seeds per size; hidden-100 is the 50-seed core; GRU at 64 and 256). (b) Median deployed loss for the two-layer stacked variant with interquartile bands: a mid-training resurgence below the initial loss, then a persistent final gap.

and 400 for the tanh controller, at hidden size 256 for the GRU, and for a two-layer stacked tanh controller with 100 units per layer, with 10 paired seeds per condition. The closed-form coupled matrix extends to all of these variants, including the stacked controller, whose linearization we validate against the autograd Jacobian. Width preserves every core signature (Figure 5a): the post-initial peak in 10/10 seeds at every size with median peak/initial ratio growing with width (10.8, 17.7, 25.9, 50.4 at hidden 50–400), final gaps below threshold, spectral stages and crossings 10/10, and crossing epochs unchanged (medians 104–105.5). The GRU keeps its final-loss divergence at hidden 256 (10/10), so the architecture contrast is not a small-GRU artifact. Depth changes the signature rather than erasing it: the stacked variant trains stably and keeps the spectral stages and crossing (10/10, median epoch 92.5), but its open-loop student converges to a persistent final gap (10/10; median 35 percent) after a mid-training resurgence that stays below its high initial loss, so the strict peak criterion does not fire (0/10; Figure 5b)—depth moves the double integrator into the GRU-like regime. These experiments probe the C1 peak/final-gap signatures and the C2/C3 spectral-stage and stability-crossing diagnostics across width and depth; we did not repeat the full targeted A1 multi-horizon tradeoff sweep at every width and depth, so scale invariance of the interval-level behavioral tradeoff remains untested.

## 9 Claim-Level Audit Matrix

Table 1, the paper's main summary, separates original claims C1–C3 from audit questions A1–A2 and attaches each to evidence, robustness and generalization results, and an actionable lesson.

## 10 Actionable Lessons for Closed-Loop RNN Studies

We propose the following checklist for closed-loop RNN papers.

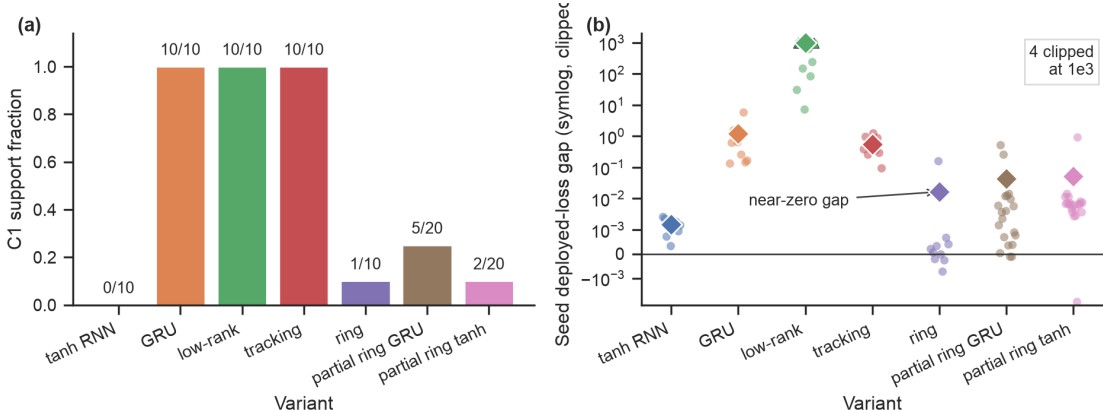

Figure 6: Generalization results across architecture and task variants. Standard variants use $n = 10$ paired seeds; the harder partial-observation ring variants use $n = 20$ paired seeds. (a) C1 final-loss support fractions with exact seed counts; tanh RNN is $0/10$ here but preserves the open-loop peak criterion in $10/10$ seeds. (b) Seed-level deployed-loss gaps with mean diamonds on a symmetric-log scale, clipped at $10^3$ for display; triangle markers indicate clipped low-rank open-loop rollout blow-ups. The simple ring is a sanity check for a nearly memoryless setting; the partial-observation ring variants are the stronger path-integration tests.

Table 1: Claim-level audit matrix for completed runs. C1–C3 are original claims under reproduction. A1–A2 are audit questions introduced here to test robustness, short/long horizon tradeoffs, and broader generalization.

| Item | Evidence type | Original protocol | A1 audit | A2 audit | Lesson |
|---|---|---|---|---|---|
| C1: open/closed divergence | Final deployed loss and peak signature | Peak 50/50; final gap 0/50 | Peak robust in finite SGD-like settings; final-gap diagnostic varies; no clipping fails | GRU/low-rank/tracking final gaps; tanh peak only; ring weak | Report deployed loss and peak signatures. |
| C2: stage structure | Spectral detector and loss-only observers | Spectral 50/50; segmented loss stages 0/50 | Spectral C2 robust in finite linearizable settings | Transfers where a (local) linearization exists | Stage claims need diagnostics in the claim's own space. |
| C3: coupled stability | Coupled spectral radius and timing | Crossing 50/50; median epoch 105.5 | Robust finite crossings | Closed form: tanh/low-rank/stacked; Jacobian crossings 60/60 for GRU/ring, near-marginal for partial-obs ring | Report the coupled spectrum and its timing vs stage boundaries. |
| A1: protocol robustness and tradeoff | Perturbation grid and multi-horizon component metric | N/A | Union/radius 120/120; loss worsening 62/120; exclusive loss-only 0/120 | N/A | Robustness claims need stress tests, explicit horizons, and component criteria. |
| A2: broader generalization | Architecture, task, scale, depth extensions | N/A | N/A | Behavioral: diagnostic-dependent; tracking strong, ring weak. Diagnostic: Sec. 8.2. Scale: signatures persist at hidden 50–400; depth: persistent gap, no strict peak | Effects depend on task structure, coupling, observability; report behavioral and diagnostic transfer separately. |

- **Report deployed closed-loop performance and peak-over-training signatures.** A model can imitate a teacher under fixed inputs yet fail when its own actions determine future inputs; final loss alone misses the transient peaks that define the phenomenon.

- **Use paired-seed comparisons when contrasting open-loop and closed-loop training:** same initialization, architecture, optimizer budget, and evaluation rollouts, changing only the training regime.

- **Define learning stages in the diagnostic space used by the claim.** Spectral stage claims should report eigenvalue criteria and boundary epochs; loss-only changepoints should be labeled as downstream observables.

- **Analyze the coupled agent-environment system** whenever a mechanistic stability claim is made: RNN-only spectra miss environment-induced stability changes, and local Jacobians extend the diagnostic beyond explicitly linearizable systems.

- **Report feedback strength, rollout horizon, per-loop gradient-clipping, and teacher-input conventions.** These choices control whether the closed-loop phenomenon appears, whether training survives at all, and whether final deployed-loss gaps persist or close.

- **Report failure rates, not just average curves, and include at least one stress test.** Means hide unstable runs and seed-dependent plateaus; perturbations are inexpensive compared with the interpretive cost of an untested mechanistic claim.

## 11  Discussion

This project treats Ger and Barak's framework as a case study in reproducibility for interactive recurrent systems. The audit supports the central trajectory-level open/closed divergence, spectral stage structure, and coupled stability transition under a main-text-aligned double-integrator protocol, and identifies limits. Final deployed loss is not the right sole diagnostic for C1: the final gap is small after open-loop recovery, while the transient peak is robust. Spectral C2 reproduces, but segmented loss-only changepoints do not recover the same ordering. A1's radius signal is necessary for loss worsening but not sufficient (58/120 radius-only runs)—a leading indicator, not a failure predictor. And generalization is hierarchical and diagnostic-dependent: some architecture and tracking variants transfer, while path-integration variants mostly do not.

These mixed results are why a claim-level audit is useful. A figure-level reproduction might only note that a curve looks similar for one seed; a methodological audit can say that final and peak diagnostics tell different stories, that missing clipping produces outright failure, that optimizer and horizon choices control whether the final gap persists, and that stage claims need spectral boundaries rather than visual annotations—which matters especially in computational neuroscience, where RNNs serve as mechanistic models and closed-loop behavior is closer to biological learning than fixed-dataset supervision.

**Scope: richer agent-environment interactions.** The coupled-stability mechanism defended here rests on three premises: a differentiable, low-dimensional environment that is locally linearizable around deployed trajectories; a dense per-step state cost; and direct gradients through the rollout. Richer agent settings break at least one: RL-style agents replace rollout gradients with high-variance likelihood-ratio estimates, multi-step decision tasks make the coupled object a long-horizon composition rather than a one-step map, and stronger partial observability forces memory to substitute for state access. Our evidence brackets these boundaries—behavioral transfer weakens under partial observation, the stacked variant probes deeper memory, and local Jacobians supply the gated-controller tooling—so we state the mechanistic claim as conditional on local linearizability and treat stochastic RL-style loops as untested. Work on global spectral signatures in other learning systems similarly prioritizes spectrum-level structure over local diagnostics (Ouyang et al., 2026).

**Limitations.** The audit does not cover every architecture, task, or full reinforcement learning; training uses direct gradients through deterministic rollouts, the path-integration extensions are lightweight, and the A1 metric is interval-level. Coupled spectral analysis, now extended to gated controllers via local Jacobians (with only marginal stabilization for the partial-observation variants), remains untested for stochastic or non-differentiable environments.

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

## A  Artifact Entry Points

The accompanying anonymous repository provides a config-driven implementation with smoke and full-run modes, raw CSV outputs, claim tables, and figure-generation scripts. The artifact is organized around the audit protocol rather than notebook-only reproduction: each experiment records the config, seed, device, metrics, and per-epoch time series. During double-blind review, the supplementary archive omits public repository URLs, Software Heritage origin links, Git remotes, commit history, local logs, and machine-specific paths; a public archival link should be added only after de-anonymization. Table 2 lists the main entry points.

## B  Metric Definitions

Let $s_t$ be the environment state, $o_t = Cs_t + \epsilon_t$ the observation, $h_t$ the recurrent state, and $u_t = f_\theta(o_t, h_t)$ the action. Closed-loop deployment follows $s_{t+1} = F(s_t, u_t)$ and $h_{t+1} = G_\theta(o_t, h_t)$. The deployed loss is

$$L_{\text{deploy}}(\theta) = \frac{1}{T} \sum_{t=0}^{T-1} \left[ \ell_s(s_t, s_t^\star) + \lambda_u \|u_t\|_2^2 \right], \tag{1}$$

Table 2: Artifact entry points. Full runtimes are approximate for one NVIDIA L40 GPU and are intended for planning rather than benchmarking.

| Component | File/script | Produces | Runtime | Expected output |
|---|---|---|---|---|
| Smoke validation | `scripts/run_smoke.sh` | Tiny raw outputs, tables, figures | < 10 min | `results/raw/smoke_*` |
| Full audit | `scripts/run_full_audit.sh` | Core, robustness, generalization, figures | 16–18 h | sweep summaries, figures |
| Targeted C2/A1/A2 | `scripts/run_targeted_c2_a1_a2.sh` | Tradeoff, hard ring, changepoints | 8–10 h | targeted summaries |
| Extended analyses | `scripts/run_revision_experiments_fast.sh` | Scale/depth, Jacobian runs, clip selectivity, convention ablation, artifact side-by-side | 2 h (2 GPUs); 21–23 h sequential | extended-analysis summaries |
| Claim tables | `make_claim_tables` module | Claim matrix | < 1 min | `claim_matrix.csv` |
| Supplemental tables | `supplemental_audit_tables` module | Run accounting, C2 sensitivity, A1 split, peak/alignment checks | < 1 min | processed CSVs |
| Figures | `make_all_figures` module | Paper figures | < 1 min | `figure_*.png` |

while teacher-forced imitation loss compares the learner action to the teacher action on teacher-generated observations:

$$L_{\text{TF}}(\theta; \theta^\star) = \frac{1}{T} \sum_{t=0}^{T-1} \| f_\theta(o_t^\star, h_t) - f_{\theta^\star}(o_t^\star, h_t^\star) \|_2^2. \tag{2}$$

For linearizable double-integrator settings we estimate an effective feedback gain by ridge regression, $\widehat{K}_t = (X^\top X + \alpha I)^{-1} X^\top U$, and report the closed/open gain distance $D_K = (\sum_{t \in \mathcal{E}} \| \widehat{K}_t^{\text{CL}} - \widehat{K}_t^{\text{OL}} \|_F^2)^{1/2}$.

For linear RNN controllers and linear environments, with $s_{t+1} = As_t + Bu_t$, $u_t = W_{ho}h_t$, and $h_{t+1} = (1-\eta)h_t + \eta(W_{hh}h_t + W_{ih}Cs_{t+1})$, the coupled transition matrix over $z_t = [s_t^\top, h_t^\top]^\top$ is

$$P_\theta = \begin{bmatrix} A & BW_{ho} \\ \eta W_{ih}CA & (1-\eta)I + \eta W_{hh} + \eta W_{ih}CBW_{ho} \end{bmatrix}. \tag{3}$$

We report $\rho_{\text{RNN}} = \max_i |\lambda_i(W_{hh})|$ and $\rho_{\text{coup}} = \max_i |\lambda_i(P_\theta)|$. The stability-crossing epoch is the first epoch at which $\rho_{\text{coup}} < 1$ for a minimum persistence window.

The open-loop peak signature is evaluated over the deployed-loss trajectory across training epochs. We count a post-initial peak when $\arg\max_e L_{\text{deploy},e}^{\text{OL}} > 0$ and $\max_e L_{\text{deploy},e}^{\text{OL}} > 1.5 L_{\text{deploy},0}^{\text{OL}}$; we also report peak/initial and peak/final ratios.

The spectral stage detector follows the original diagnostic structure. It records Stage 1 support when the effective position gain is negative and the coupled system has a persistent unstable complex mode; Stage 2 support when the dominant coupled radius persistently enters the unit disk; and Stage 3 support when the largest real coupled eigenvalue grows after that stability crossing. For comparison, loss-only labels use smoothed log deployed loss $y_t = \log(\max(L_{\text{deploy},t}, \varepsilon))$. The segmented loss variant fits three piecewise-linear segments and requires Stage 1 slope $< -0.02$, Stage 2 slope magnitude below $\max(0.35|\text{slope}_1|, 0.004)$, and Stage 3 slope $< \text{slope}_2 - 0.002$.

For A1, define a myopic-improvement interval by $\Delta \log L_e^{(T_s)} < -\delta_{\text{short}}$. We report four component criteria: union $I_s \wedge (I_\ell \vee I_\rho)$, loss $I_s \wedge I_\ell$, radius $I_s \wedge I_\rho$, and both $I_s \wedge I_\ell \wedge I_\rho$, where $I_\ell$ is $\Delta \log L_e^{(T_\ell)} > \delta_{\text{long}}$ and $I_\rho$ is $\Delta \rho_{\text{coup},e} > 0$. In the targeted sweep, $T_s = 10$, $T_\ell = 200$, and $\delta_{\text{short}} = \delta_{\text{long}} = 0.005$.

For scalar paired metric $M$, the open/closed effect is $\Delta_i(M) = M_i^{\text{OL}} - M_i^{\text{CL}}$. We report mean, nonparametric bootstrap 95 percent CI, seed-level SD/IQR where useful, direction agreement $N^{-1} \sum_i \mathbf{1}[\Delta_i(M) > 0]$, and binary claim-support rate $S(c) = N^{-1} \sum_i q_i(c)$.

For controllers without an explicit recurrent matrix (GRUs) and tasks without a linear state-space form (ring variants), we replace the closed-form coupled matrix with a local Jacobian: at probe points $(s_t, h_t)$ collected along deployed rollouts, we differentiate the one-step coupled map $f : (s_t, h_t) \mapsto (s_{t+1}, h_{t+1})$ (with parameters frozen and observation noise disabled) and report the spectral summary of $\partial f / \partial (s, h)$, aggregated as the median over probes. Where both constructions apply, their spectral radii agree

at the origin to within $10^{-5}$ (Appendix I). Local-Jacobian stability crossings carry the separate flag `claim_C3_stability_transition_jacobian`.

## C   Audit Protocol Specification

Table 3: Direct mapping from original evidence to our reproduction tests. This separates figure-level reproduction from claim-level decisions.

| Original claim | Original evidence | Our test | Result | Decision |
|---|---|---|---|---|
| Closed/open divergence | Loss/gain figures and open-loop sharp peak | 50 paired seeds plus peak-signature check | post-initial peak 50/50; final gap 0/50 | Reproduced trajectory signature |
| Stage structure | Spectral stages: negative-position policy, stability phase, refinement | Spectral stage detector plus loss-only changepoints | spectral stages 50/50; segmented loss stages 0/50 | Reproduced spectrally |
| Coupled stability | Coupled eigenvalue argument and Stage 2 alignment | Coupled radius crossing and spectral stage timing | 50/50 crossings; median crossing 105.5 | Reproduced |
| Motor-control applicability | Tracking extension | Tracking plus ring variants | tracking 10/10; ring-family 8/50 | Partial boundary |

Table 4: Robustness perturbation grid. The original protocol also has the separate 50-seed core reproduction; the robustness grid repeats the baseline for 15 additional paired seeds so every perturbation has the same seed count.

| Perturbation | Changed value | Seeds |
|---|---|---|
| `original` | Baseline double-integrator protocol | 50 core; 15 grid |
| `lr_low` | learning rate 0.003 | 15 |
| `lr_high` | learning rate 0.03 | 15 |
| `init_weak` | recurrent initialization scale 0.03 | 15 |
| `init_strong` | recurrent initialization scale 0.3 | 15 |
| `feedback_low` | environment feedback strength 0.5 | 15 |
| `feedback_high` | environment feedback strength 1.5 | 15 |
| `episode_short` | rollout horizon 25 steps | 15 |
| `episode_long` | rollout horizon 100 steps | 15 |
| `obs_noise` | observation noise standard deviation 0.05 | 15 |
| `adam` | optimizer Adam with learning rate 0.001 | 15 |
| `high_control_penalty` | control penalty 0.05 | 15 |
| `no_clip` | gradient clipping disabled | 15 |

## D   Protocol Provenance

Table 6 classifies every protocol setting into three tiers: *exact* settings match the public artifact; *main-text-aligned* settings deviate from the public nonlinear notebook to follow the paper's main text; *audit choices* are new to this study and were fixed during smoke-test development before the reported sweeps. The full machine-generated map is `results/processed/provenance_map.csv`. The 22 exact-tier settings comprise the task dynamics ($dt$, feedback strength, zero observation noise), the model (tanh RNN, hidden size 100, initialization scales 0.1, leak 1), and training (1001 epochs, batch 100, 50-step rollouts, SGD at learning rate 0.01, gradient clip 1.0, zero initial action, before-step loss timing, summed control cost, white-noise open-loop inputs).

## E   Detector Threshold Sensitivity

Table 7 varies the final-gap threshold across 1–20 percent. Every setting keeps its claim-level classification at and above the pre-specified 5 percent value: the four robustness settings with full support at 5 percent

Table 5: Closed-loop RNN reproducibility audit protocol. The protocol is intentionally lightweight: each row defines a required audit step, its minimum implementation, and the output needed to support claim-level conclusions.

| Audit step | Minimum requirement | Recorded output |
| --- | --- | --- |
| Paired training | Clone the same initialization into open-loop and closed-loop learners; keep architecture and evaluation budget fixed. | Per-seed train loss, deployed closed-loop test loss, peak-over-training deployed loss, final loss, and effective-gain trajectory for both learners. |
| Seed-level uncertainty | Run enough seeds to estimate direction agreement and confidence intervals; report seed count explicitly. | Mean, 95 percent bootstrap CI, fraction of seeds supporting each claim, and seed-level failure/recovery rates. |
| Stage detection | Predefine detectors in the same diagnostic space as the claim; for this audit, separate loss-only stages from spectral stages. | Stage labels, plateau duration, transition epochs, spectral-crossing gaps, and frequency of detected stages across seeds/settings. |
| Coupled diagnostics | Compute RNN-only and coupled agent-environment spectra whenever the system can be linearized or locally linearized. | Spectral radius time series, stability-crossing epoch, plateau-exit gap, and predictive correlation with recovery/failure. |
| Robustness perturbations | Change factors that alter optimization or environment coupling, such as optimizer, initialization scale, feedback strength, noise, horizon, control penalty, and clipping. | Claim support under each perturbation, effect-size changes, and perturbation-level failure rates. |
| Generalization check | Test at least one architecture or task variant that preserves action-dependent inputs and one that changes the recurrent/control structure. | Claim support by variant and a boundary-condition statement explaining when the phenomenon does or does not transfer. |
| Claim/audit decision | Classify each claim or audit question as reproduced, partially reproduced, or not reproduced using the same criteria across settings. | Matrix linking evidence type, original setting, robustness, generalization, and actionable lesson. |

Table 6: Protocol provenance: main-text-aligned deviations and audit choices. The 22 exact artifact matches are listed in the text above; the machine-generated map with per-setting notes is `provenance_map.csv`.

| Setting | Public notebook | This audit |
| --- | --- | --- |
| *Main-text-aligned (5)* | | |
| initial state, lower bound | $-1$ | $-2$ (main text) |
| initial state, upper bound | 1 | 2 (main text) |
| control penalty $\beta$ | 0.005 | 0 (main text) |
| state-cost reduction | mean over elements | squared Euclidean state cost, batch mean |
| loss normalization | summed over rollout | averaged over rollout $(1/T)$ |
| *Audit choices (10); original does not specify* | | |
| final-gap threshold | — | relative deployed-loss gap $> 5$ percent |
| peak criterion | — | peak $> 1.5\times$ initial, epoch $> 0$ |
| spike criterion | — | peak $> 2.0 \times \max(\text{initial}, \text{final})$ |
| spectral stage detector | — | gain $< -0.01$, persistence 5, $\lambda_3$ growth $> 0.01$ |
| changepoint criteria | — | slopes $-0.02$ / $0.35\times$ / $-0.002$; min segment 20, stride 2 |
| A1 horizons and $\delta$ | — | $[10, 50, 200]$; $\delta = 0.005$ |
| seed counts | — | 50 core / 15 robustness / 20 tradeoff / 10 generalization |
| evaluation protocol | — | fixed 10-episode eval batch, eval seed 12345 |
| perturbation grid | — | Table 4 values |
| ridge gain $\alpha$ | — | $10^{-8}$ in effective-gain regression |

(`adam`, `episode_short`, `lr_high`, `lr_low`) remain 15/15 across the entire band, the core setting remains 0/50 everywhere, and GRU, low-rank, and tracking variants keep their support. At the loosest 1–2 percent thresholds, small sub-majority counts appear in a few perturbations (`feedback_low` 3/15, `obs_noise` 3/15,

`init_strong` 1/15 at 1 percent), and the partial-observation tanh ring changes qualitatively (19/20 at 1 percent, 13/20 at 2 percent, 2/20 at 5 percent): its deployed-loss gaps are real but sit below the pre-specified smallest effect size of interest, and we report this boundary case explicitly.

Table 7: C1 final-gap support counts as the relative deployed-loss threshold varies. The value pre-specified before the full reported sweeps is 5 percent.

| Setting | $n$ | 1% | 2% | 5% | 10% | 20% |
|---|---|---|---|---|---|---|
| original (core) | 50 | 0 | 0 | 0 | 0 | 0 |
| original (robustness grid) | 15 | 0 | 0 | 0 | 0 | 0 |
| adam / episode_short / lr_high / lr_low | 15 each | 15 | 15 | 15 | 15 | 15 |
| feedback_low | 15 | 3 | 0 | 0 | 0 | 0 |
| obs_noise | 15 | 3 | 2 | 0 | 0 | 0 |
| init_strong | 15 | 1 | 0 | 0 | 0 | 0 |
| init_weak / feedback_high / episode_long / high_control_penalty | 15 each | 0 | 0 | 0 | 0 | 0 |
| no_clip (all runs non-finite) | 15 | 0 | 0 | 0 | 0 | 0 |
| gru / low_rank | 10 each | 10 | 10 | 10 | 10 | 10 |
| tracking_task | 10 | 10 | 10 | 10 | 10 | 9 |
| tanh_rnn | 10 | 0 | 0 | 0 | 0 | 0 |
| ring_path_integration | 10 | 1 | 1 | 1 | 1 | 1 |
| ring_partial_obs_gru | 20 | 10 | 8 | 5 | 2 | 2 |
| ring_partial_obs_tanh | 20 | 19 | 13 | 2 | 1 | 1 |

For A1, we recompute the four component criteria over $\delta \in \{0.0025, 0.005, 0.01, 0.02\}$ and horizon pairs $\{(10, 200), (10, 50), (50, 200)\}$ from the per-epoch multi-horizon series. The union criterion and the radius criterion are supported in 120/120 runs, and the exclusive loss-only criterion in 0/120, in *every* cell of the grid; only the count of runs whose loss-worsening criterion fires moves with the thresholds (Table 8). That count is threshold-dependent in both directions—small $\delta$ inflates the improvement-interval denominator so the conditional-fraction gate fails, and large $\delta$ suppresses qualifying steps—which is exactly why the grid-invariant union and radius criteria, not the loss count, carry the A1 claim.

Table 8: A1 sensitivity grid. Cell entries are runs (of 120) whose long-horizon loss-worsening criterion fires ($\geq 3$ qualifying steps and conditional fraction $\geq 0.1$). In every cell the union and radius criteria are 120/120 and the exclusive loss-only criterion is 0/120. The setting pre-specified before the full reported sweeps is $\delta = 0.005$ with horizons $(10, 200)$.

| $\delta$ | $(T_s, T_\ell) = (10, 200)$ | $(50, 200)$ | $(10, 50)$ |
|---|---|---|---|
| 0.0025 | 2 | 0 | 0 |
| 0.005 | **62** | 29 | 0 |
| 0.01 | 107 | 99 | 0 |
| 0.02 | 117 | 25 | 0 |

Table 9 reports the corrected stage-sensitivity analysis. The original version of this table was degenerate: the five threshold variants relabel a fixed least-squares segmentation, so boundary and slope columns were identical by construction. The corrected analysis adds variants that perturb the segmentation itself (stride 1, minimum segment 40, smoothed log-loss input) and reports component-level support. The fitted boundaries are nearly invariant across all eight variants ($\tau_1$ 110–120, $\tau_2$ 192–194, matching the Section 7 medians), and the component supports show the failure is structural: at the main thresholds even the stage-1 fast-descent label does not fire (fitted first-segment slope $-0.0117$ against the $-0.02$ cutoff; it fires only under loosened slopes, 47–50/50), no variant detects the stage-2 slow phase or stage-3 reacceleration, and strict three-stage support is 0/50 everywhere.

## F   Gradient-Clipping Failure Analysis

**Forensics.** In all 15 unclipped runs (`no_clip_refresh`, reproducing the original `no_clip` failure 15/15) the closed-loop learner fails first and fastest: its training loss is non-finite by epoch 5–7 and its deployed loss by epoch 4–6, with the coupled spectral radius growing without bound from an initial value at or marginally above the stability boundary (1.000–1.045 across seeds). Because the diverged closed-loop model becomes

Table 9: C2 sensitivity of the segmented loss-only changepoint observer on the 50 original-setting seeds, corrected to include segmentation-perturbing variants and component-level support. The spectral C2 detector is supported in 50/50 seeds throughout; this loss-only observer never recovers the strict three-stage ordering.

| Detector variant | Stage-1 fast | Strict three-stage | Median $\tau_1$ | Median $\tau_2$ |
|---|---|---|---|---|
| Main segmented | 0/50 | 0/50 | 120 | 193 |
| Loose segmented | 47/50 | 0/50 | 120 | 193 |
| Strict segmented | 0/50 | 0/50 | 120 | 193 |
| Short-segment binary style | 0/50 | 0/50 | 120 | 193 |
| Very loose reacceleration | 50/50 | 0/50 | 120 | 193 |
| Stride-1 segmentation | 0/50 | 0/50 | 119 | 192.5 |
| Min-segment-40 segmentation | 0/50 | 0/50 | 110 | 194 |
| Smoothed log-loss (window 9) | 0/50 | 0/50 | 118 | 192 |

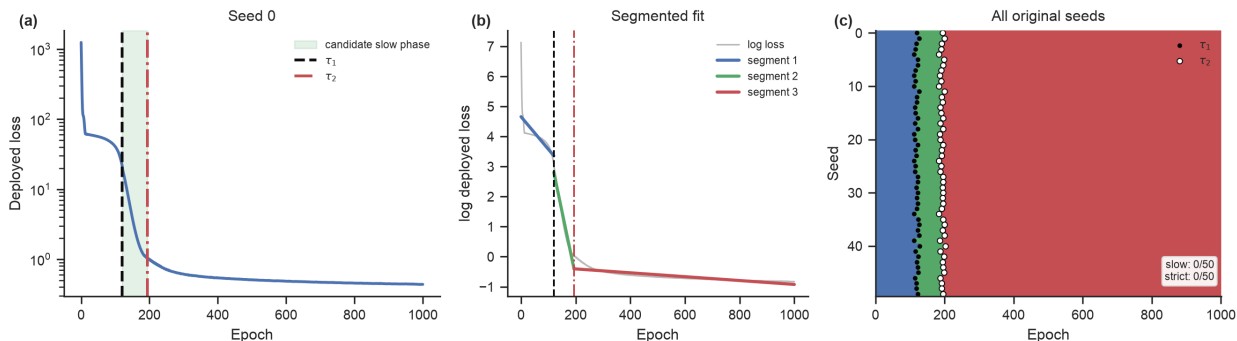

Figure 7: Stage analysis for the original-setting runs using downstream loss changepoints. (a,b) A representative seed with candidate boundaries $\tau_1$ and $\tau_2$ and segmented fits. (c) Aggregate stage raster for all 50 original seeds; black dots mark $\tau_1$ and white dots mark $\tau_2$. The segmented detector uses minimum segment length 20 epochs, Stage 1 slope $< -0.02$, Stage 2 slope magnitude below $\max(0.35|\text{slope}_1|, 0.004)$, and Stage 3 reacceleration slope $< \text{slope}_2 - 0.002$. Unlike the spectral detector, this loss-only segmented observer does not recover robust strict three-stage support.

the imitation teacher, the open-loop student's training signal is non-finite from its first epoch; the open loop never gets the chance to fail on its own. Per-seed details are in `no_clip_forensics.csv`.

**Selectivity** (15 seeds per condition). `no_clip_closed_only` (closed loop unclipped, open loop clipped): 0/15 finite. `no_clip_open_only` (closed loop clipped, open loop unclipped): 15/15 finite with the peak-and-recovery signature in 15/15—the imitation loop does not need clipping at all. `no_clip_adam` (fully unclipped, Adam at learning rate 0.001): 15/15 finite, with the optimizer-specific behavior of the main-grid Adam row (peak in 6/15). `no_clip_lr_low` (fully unclipped, SGD at learning rate 0.003): 0/15 finite. Clipping the closed-loop learner alone is necessary and sufficient in the SGD regime, and adaptivity, not step size, is what rescues the unclipped condition.

## G   Open-Loop Teacher-Input Convention Ablation

In 15 paired seeds trained identically except for the open-loop input convention, teacher-rollout inputs produce a persistent final deployed-loss gap in 15/15 seeds (median relative gap $\approx 147$, i.e. the deployed open-loop loss remains two orders of magnitude above the closed-loop loss) with no post-initial peak in any seed, while the white-noise convention shows the peak followed by final convergence in all 65 comparable runs (50 core seeds plus the 15-seed robustness baseline), with every final gap below the 5 percent threshold. Per-seed values are in the `robustness_open_loop_teacher_rollout` summary.

## H  Original-Artifact Side-by-Side Comparison

Table 10 compares seeded executions of the original `train_non_linear` notebook (5 seeds; the notebook sets no seed, so we inject one) against package runs under the artifact-protocol configuration (10 seeds). All compared quantities agree within seed variability; notably, the original notebook itself reproduces the C1 reframing—a large transient open-loop peak followed by final convergence (notebook final gap 0.0005, below 1 percent). Coupled-radius metrics exist on the package side only, since the notebook does not compute spectra.

Table 10: Original-artifact notebook versus package replication under the notebook protocol. Mean [min, max] over seeds.

| Metric | Notebook (5 seeds) | Package (10 seeds) |
|---|---|---|
| Closed final deployed loss | 0.061 [0.047, 0.073] | 0.059 [0.058, 0.060] |
| Open final deployed loss | 0.062 [0.047, 0.075] | 0.060 [0.059, 0.061] |
| Open-loop peak epoch | 213.2 [212, 214] | 212.2 [202, 218] |
| Open peak/initial ratio | 47.9 [27.3, 80.0] | 39.8 [15.5, 68.9] |
| First coupled-stable epoch | — | 103.1 [98, 112] |

## I  Local-Jacobian Validation and Diagnostic-Transfer Trajectories

**Validation.** On linear and tanh double-integrator systems the Jacobian spectral radius at the origin matches the closed-form coupled radius to within $10^{-5}$ (enforced in the test suite, including the two-layer stacked closed form against the autograd Jacobian). On 10 core-protocol seeds instrumented with both constructions, the along-trajectory Jacobian radius tracks the zero-point closed form with median absolute difference $3.6 \times 10^{-7}$, maximum $5 \times 10^{-3}$, and correlation 1.00 (Figure 8e).

**Diagnostic-transfer trajectories.** Figure 8 shows median local-Jacobian coupled-radius trajectories with per-seed crossing ticks. Crossings occur early and stabilization is decisive for the GRU double integrator (median crossing epoch 10, final radius 0.58) and the simple ring (epoch 5, final 0.83); the partial-observation ring variants cross late or marginally (median epochs 40 and 270) and hover near the boundary throughout (final radii 0.986 and 0.978, with individual seeds up to 1.19), matching their weak behavioral transfer.

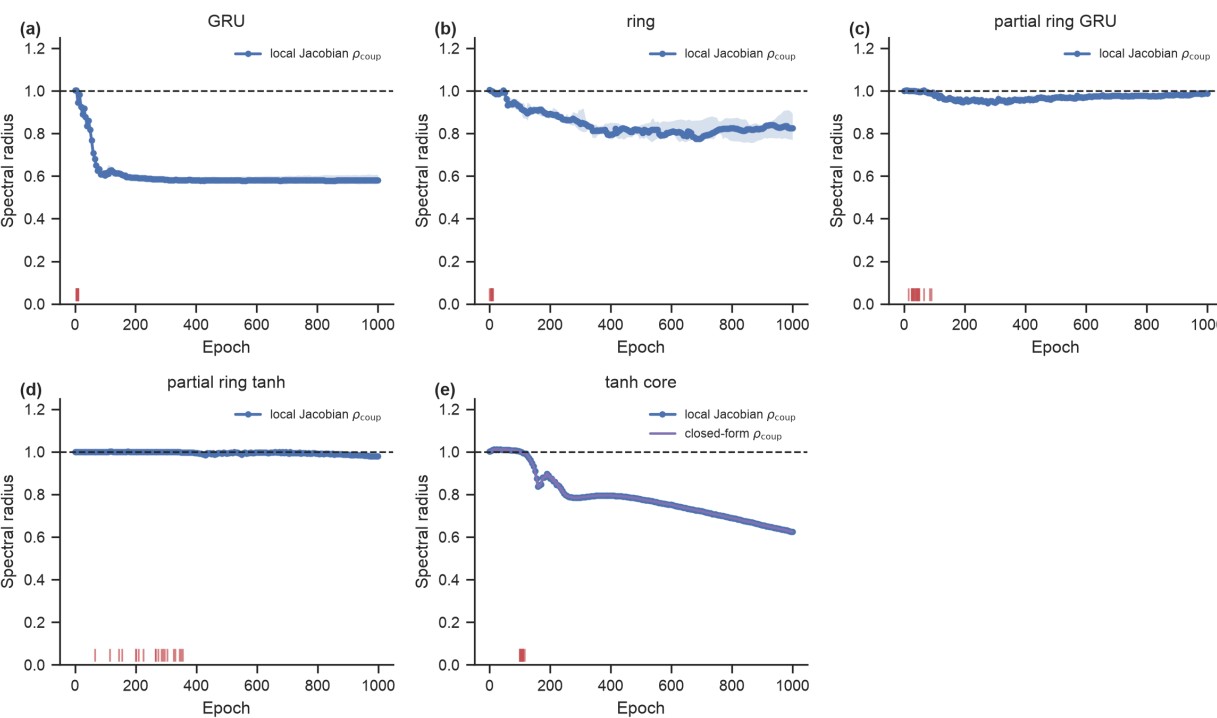

Figure 8: Local-Jacobian coupled spectral radius over training for the diagnostic-transfer runs (median with interquartile band over seeds; probes every 5 epochs; dashed line at $\rho = 1$; ticks mark per-seed crossing epochs). (a) GRU double integrator. (b) Ring path integration. (c,d) Partial-observation ring with GRU and tanh controllers. (e) tanh core protocol, where the along-trajectory Jacobian radius overlaps the zero-point closed-form radius.

