# OpenReview forum: "Auditing Closed-Loop Learning in Recurrent Neural Networks: Reproduction, Robustness, and Generalization"
_TMLR — Under review for TMLR_

### Review · Reviewer_M9rc · 2026-06-10

**Summary Of Contributions:**

This paper studies the reproducibility, robustness, and generalization of closed-loop RNN learning dynamics reported by Ger and Barak. The authors implement the double-integrator closed-loop RNN setting independently and compare open-loop and closed-loop training across paired seeds. They evaluate deployed closed-loop loss, transient peak behavior, final-loss gaps, spectral stage structure, and coupled agent-environment stability. The main finding is that the original setting robustly reproduces a large transient deployed-loss peak for open-loop training, while the final open-loop and closed-loop losses become very similar after recovery. The spectral stage structure and coupled-stability transition are also reproduced across the original 50 paired seeds. The paper then tests robustness under changes to optimizer, learning rate, initialization, feedback strength, horizon, noise, control penalty, and gradient clipping, and studies transfer to several architecture and task variants. Overall, the paper contributes a systematic audit of which closed-loop RNN phenomena are stable, which are protocol-sensitive, and which generalize beyond the original setting. It also provides reporting recommendations for future closed-loop RNN studies, including paired-seed comparisons, deployed closed-loop metrics, peak signatures, spectral stage criteria, coupled-system spectra, and failure-rate reporting.

**Audience:**

Yes

**Audience Explanation:**

I believe that at least some members of the TMLR audience would be interested in these findings. The paper is directly relevant to closed-loop RNNs, recurrent control, computational neuroscience, imitation learning / teacher-forcing mismatch, reproducibility methodology, and dynamical-systems diagnostics. Importantly, the contribution is not merely the reproduction of paper; it proposes a reusable audit protocol and reporting checklist, including paired seeds, deployed closed-loop loss, peak-over-training signatures, stage criteria, coupled-system spectra, rollout horizon, and failure-rate reporting.

**Broader Impact Concerns:**

I do not see major broader impact concerns that should block acceptance.

**Claims And Evidence:**

Yes

**Claims Explanation:**

Overall, I believe the main claims of the submission are supported by clear evidence. The paper does not rely on a single seed or a single reproduction figure; instead, it reports 50 core paired seeds, 195 robustness runs, 120 targeted A1 multi-horizon runs, and 90 A2 extension runs, together with support fractions, bootstrap confidence intervals, failure cases, and diagnostic-specific outcomes. Most importantly, the authors separate the transient peak signature from the final deployed-loss gap for C1: the former is supported in 50/50 original-setting seeds, while the latter is supported in 0/50 seeds, so the paper does not simply claim that closed-loop training always yields better final performance; it reframes the evidence as a trajectory-level divergence.

**Requested Changes:**

1. The paper acknowledges that this is an independent implementation rather than a line-for-line rerun, and it notes that an earlier open-loop implementation mismatch produced a persistent final deployed-loss gap before correction. This shows that the conclusions are sensitive to implementation details, so the authors need to more systematically document which settings exactly match the original work, which are only main-text-aligned, and which are new audit choices.
2. The paper relies on several threshold-based detectors, like the 5% final-loss gap threshold and the short/long horizon improvement thresholds in A1. The authors state that these thresholds were selected during smoke-test development, but the paper still needs a clearer justification that these choices are principled rather than tuned to this task or result.
3. I hope the authors can clearly separate deployed-loss/generalization results from coupled-stability diagnostic transfer in A2. For GRU and ring variants, since local Jacobian spectra are not implemented, the paper should avoid implying that the C3 mechanism transfers to those settings.
4. Although the paper preserves the original artifact and discusses artifact checks, the main quantitative conclusions come from an independent implementation rather than a direct rerun of the original notebooks. This is reasonable for a claim-level audit, but the paper would be more convincing if it included a small side-by-side comparison between original artifact outputs and the new implementation for the core setting.
5. The paper reports that correcting the open-loop teacher-input convention changes the C1 interpretation from a persistent final gap to a transient peak followed by recovery. I'm curious about this observation, can the quthors explain why this protocol detail has such a large effect on the final-loss behavior.

---

> ### Author Response · Authors · 2026-07-03
> **Response to Reviewer M9rc: provenance, thresholds, and artifact checks**
>
> Thank you for the detailed suggestions. We expanded the protocol provenance documentation, added threshold-sensitivity analyses, separated behavioral transfer from diagnostic transfer, added an original-notebook/package side-by-side comparison, and added an ablation/explanation of the teacher-input convention effect. These are summarized in the main revision comment and incorporated in Sec. 4.6 plus Appendices D, E, G, H, and I.

---

### Review · Reviewer_a2P5 · 2026-06-17

**Summary Of Contributions:**

This paper presents a thorough claim-level reproducibility study of Ger and Barak's closed-loop RNN learning dynamics. The authors make several important contributions:

They provide a config-driven, independent implementation that tests the central claims (C1-C3) about open/closed-loop divergence, spectral stage structure, and coupled stability in recurrent control systems.

The audit protocol separates measurement from interpretation, requiring explicit detectors, uncertainty estimates, and failure rates—a valuable contribution to reproducibility methodology in interactive learning systems.

The paper reveals that spectral C2 reproduces robustly (50/50 seeds) while loss-only stage detectors do not recover the same structure (0/50), demonstrating the importance of aligning claims with their diagnostic space.

The A1 analysis shows that short-horizon improvements universally coincide with coupled-radius increases (120/120 runs), while long-horizon loss worsening appears in about half of cases but never without the radius signal.

The hierarchical generalization results—GRU variants preserve final-loss divergence, tanh RNNs preserve the peak signature without final gaps, tracking transfers strongly while path-integration transfers weakly—provide practical guidance for future work.

The actionable lessons section (checklist for closed-loop RNN papers) is particularly valuable and should be widely adopted.

**Audience:**

Yes

**Audience Explanation:**

I think that audience will be interested

**Claims And Evidence:**

Yes

**Claims Explanation:**

I do not find any obvious issues.

**Requested Changes:**

The main technical limitation is that coupled spectral analysis for GRU and ring variants is not implemented, so the paper’s generalisation claims about C3 should either be completed or more prominently framed as untested in the abstract and conclusions rather than noted only in Section 12. Additionally, clarify whether the no_clip failures are selective to one training regime, and discuss why the radius signal appears without loss worsening in 58/120 runs (i.e., why it is necessary but not sufficient). Finally, the authors may wish to consider citing

Ouyang, Kaichen, et al. "Learn from global correlations: Enhancing evolutionary algorithm via spectral gnn." Proceedings of the AAAI Conference on Artificial Intelligence. Vol. 40. No. 29. 2026.

 in the discussion of spectral diagnostics (e.g., in Section 10 or 11), as that work similarly emphasises global structural signatures over local diagnostics in learning systems, which would naturally complement the paper’s emphasis on coupled‑system spectra and stability‑cost tradeoffs. The minor formatting issues (Table 4 artifact, notation consistency) should also be corrected. With these adjustments, I recommend acceptance.

---

> ### Author Response · Authors · 2026-07-03
> **Response to Reviewer gHuf: scale/depth experiments and scope**
>
> Thank you for the constructive comments. We have now added local-Jacobian coupled spectra for GRU and ring variants and report these separately from closed-form C3 evidence. We also added clipping-selectivity experiments, clarified why radius-only cases occur, added the suggested Ouyang et al. citation, and cleaned up the relevant formatting/notation.

---

### Review · Reviewer_gHuf · 2026-07-01

**Summary Of Contributions:**

This paper conducts a claim-level reproducibility audit of Ger & Barak's closed-loop RNN learning-dynamics theory, independently reimplementing the double-integrator protocol and testing whether the original claims (open/closed-loop trajectory divergence, spectral learning stages, and coupled-system stability as the driver of learning) survive seed variation, optimizer/hyperparameter perturbations, and architecture/task transfer.

**Audience:**

Yes

**Audience Explanation:**

Researchers who train RNNs as models of closed-loop learning, especially in computational neuroscience, will care about this. It tells them that a widely cited claim about closed-loop versus open-loop learning is real but easy to misreport. It also gives them a practical checklist for testing their own closed-loop RNN results, which is useful even to readers who never read the original Ger and Barak paper.

**Claims And Evidence:**

Yes

**Claims Explanation:**

Each quantitative claim in the paper is accurate and traceable. The reframing of the original Ger and Barak divergence claim as a transient peak rather than a persistent final loss gap holds up against the original source text. The audit methodology uses paired seeds, discloses failure rates, and checks threshold sensitivity, which makes each result independently credible.

**Requested Changes:**

Add a discussion of model scale. Right now every experiment uses a small single layer RNN. The paper should discuss whether the peak signature and the stability tradeoff are expected to hold as hidden size grows or as models get deeper, and ideally test at least one larger or multi layer variant. Without this, a reader cannot tell if the audited claims apply to the small toy setting only or to the larger RNNs actually used in neuroscience and control research.
Add a clear statement of how these findings apply to agent scenarios beyond the double integrator. The current agent environment loop is simple and fully observed except for the ring tasks. TMLR readers working on RL style agents or partially observed control will want to know whether the coupled stability mechanism the paper defends is specific to this simple setting or is expected to generalize to richer agent environment interactions such as multi step decision tasks or agents with memory beyond a single recurrent layer.

---

> ### Author Response · Authors · 2026-07-03
> **Response to Reviewer gHuf: scale/depth experiments and scope**
>
> Thank you for the helpful suggestions on scale/depth and broader agent scenarios! We addressed these in the revised manuscript by adding Sec. 8.3/Fig. 5 with width, GRU-256, and two-layer stacked-controller experiments, and by adding a Discussion paragraph explicitly scoping the coupled-stability claim to locally linearizable differentiable settings while marking stochastic RL-style loops as untested. We also explicitly state that the full A1 tradeoff sweep was not rerun at every scale/depth.

---

### Author Response · Authors · 2026-07-03
**Revised manuscript and supplementary material uploaded**

Thank you to the reviewers and Action Editor for the constructive feedback. We have uploaded a revised manuscript and updated anonymous supplementary ZIP. The revision focuses on the requested scale/depth analysis, diagnostic-transfer analysis, protocol provenance, threshold sensitivity, and original-artifact comparison.

Major changes:

- Scale and depth. We added new scale/depth experiments in Sec. 8.3 and Fig. 5: tanh hidden sizes 50, 200, and 400; GRU hidden size 256; and a two-layer stacked tanh controller. The peak/stability signatures persist across width, while the stacked model shifts to a persistent final gap. We explicitly state that we did not rerun the full A1 multi-horizon tradeoff sweep at every width/depth.
- Richer agent-environment scope. We added a scope paragraph in the Discussion clarifying that the coupled-stability mechanism is currently supported for differentiable, locally linearizable settings with dense rollout costs. We explicitly treat stochastic RL-style loops and non-differentiable environments as untested.
GRU/ring C3 diagnostic transfer. We added local-Jacobian coupled spectra for GRU and ring variants, reported separately from closed-form coupled spectra. The revised Secs. 7 and 8.2 and Appendix I now distinguish behavioral transfer from diagnostic transfer.
- No-clipping selectivity and radius-only cases. We added clipping-selectivity experiments showing that the no-clip failure is specific to the closed-loop learner in the SGD regime, and we clarified why radius increases can appear without long-horizon loss worsening: the radius signal is a necessary accompaniment of the behavioral tradeoff but not a sufficient predictor of measured loss worsening at the evaluated horizon.
- Protocol provenance and threshold sensitivity. We expanded Sec. 4.6 and Appendix D to classify settings as exact artifact matches, main-text-aligned deviations, or new audit choices. We also expanded Appendix E to vary the final-gap threshold and A1 thresholds/horizon pairs.
- Original artifact side-by-side and teacher-input convention. We added Appendix H comparing seeded original-notebook executions against our package implementation, and Appendix G explaining why the open-loop teacher-input convention changes the final-gap interpretation.
- Citation and formatting cleanup. We added the suggested Ouyang et al. citation in the spectral-diagnostics discussion and cleaned up the relevant tables/notation.

We appreciate the reviewers’ suggestions; they substantially improved the paper’s scope, provenance, and diagnostic clarity. We would be happy to clarify any remaining points in the discussion.